# Optimization of Thymidine Kinase-Based Safety Switch for Neural Cell Therapy

**DOI:** 10.3390/cells11030502

**Published:** 2022-01-31

**Authors:** Manon Locatelli, Flavien Delhaes, Ophélie Cherpin, Margaret E. Black, Stéphanie Carnesecchi, Olivier Preynat-Seauve, Youssef Hibaoui, Karl-Heinz Krause

**Affiliations:** 1Department of Pathology and Immunology, Faculty of Medicine, University of Geneva, CH-1211 Geneva, Switzerland; Flavien.Delhaes@unige.ch (F.D.); Ophelie.Cherpin@unige.ch (O.C.); Stephanie.CarnesecchiAcker@unige.ch (S.C.); Youssef.Hibaoui@unige.ch (Y.H.); 2School of Molecular Biosciences, Washington State University, Pullman, WA 99163, USA; blackm@wsu.edu; 3Laboratory of Therapy and Stem Cells, Geneva University Hospitals, CH-1211 Geneva, Switzerland; Olivier.Preynat-Seauve@unige.ch; 4Department of Medicine, Faculty of Medicine, University of Geneva, CH-1211 Geneva, Switzerland; 5Department of Genetic Medicine and Laboratory, Geneva Hospitals, CH-1211 Geneva, Switzerland

**Keywords:** SR39, penciclovir, pluripotent stem cell therapy, teratoma, post-transplantation tumor

## Abstract

Cell therapies based on pluripotent stem cells (PSC), have opened new therapeutic strategies for neurodegenerative diseases. However, insufficiently differentiated PSC can lead to tumor formation. Ideally, safety switch therapies should selectively kill proliferative transplant cells while preserving post-mitotic neurons. In this study, we evaluated the potential of nucleoside analogs and thymidine kinase-based suicide genes. Among tested thymidine kinase variants, the humanized SR39 (SR39h) variant rendered cells most sensitive to suicide induction. Unexpectedly, post-mitotic neurons with ubiquitous SR39h expression were killed by ganciclovir, but were spared when SR39h was expressed under the control of the cell cycle-dependent Ki67 promoter. The efficacy of six different nucleoside analogs to induce cell death was then evaluated. Penciclovir (PCV) showed the most interesting properties with an efficiency comparable to ganciclovir (GCV), but low toxicity. We tested three nucleoside analogs in vivo: at concentrations of 40 mg/kg/day, PCV and GCV prevented tumor formation, while acyclovir (ACV) did not. In summary, SR39h under the control of a cell cycle-dependent promoter appears most efficient and selective as safety switch for neural transplants. In this setting, PCV and GCV are efficient inducers of cell death. Because of its low toxicity, PCV might become a preferred alternative to GCV.

## 1. Introduction

The increase in human lifespan is associated with the predominance of age-related diseases, in particular neurodegenerative disorders, such as Parkinson’s disease. The latter is characterized by the loss of dopaminergic neurons [1]. Cell therapy approaches hold the potential of curative treatment through transplantation of therapeutic cells. The clinical results of the transplantation of fetal neurons in Parkinson patient have shown promise [2]; however, this non-sustainable source of dopaminergic neurons raises logistical and ethical problems.

Pluripotent stem cells (PSC) have great potential in tissue regeneration and repair based on their capacity for unlimited self-renewal and their ability to differentiate into virtually all cell types [3]. However, this unique feature of unlimited cell division may also lead to tumor formation and represents the major limitation for their clinical use. Indeed, even using optimal differentiation protocols, some dividing cells often persist. The most promising approach to overcome this issue consists in a safety switch, i.e., transgenic insertion of a suicide gene allowing elimination of transplanted cells in case of tumor formation. However, in its crudest form, safety switches for neural cell therapy is “throwing the baby out with the bathwater”, as both therapeutic cells and tumor forming cells would be eliminated. We previously developed a thymidine kinase (TK)-based approach, where the suicide gene is expressed under the control of the cell cycle-dependent Ki67 promoter [4]. This allows selective removal of proliferating and thus potentially tumor-forming cells, while sparing the post-mitotic therapeutic neurons. The use of a cell cycle-dependent promoter provides a second advantage: the potential immune reaction against the microbial antigen HSV-TK will be directed only against proliferating, but not against post-mitotic mature cells.

TK-based suicide gene approaches have been extensively studied as cancer therapy [5]. Unlike human thymidine kinase, the thymidine kinase of Herpes Simplex Virus type 1 (HSV-TK) phosphorylates a broad range of pyrimidine and guanosine analogs, including the antivirals acyclovir (ACV), ganciclovir (GCV), penciclovir (PCV), and brivudine (BRVD) [6]. Inside the cell, these antiviral guanosine analogs are initially phosphorylated by HSV-TK to form a monophosphate, and subsequently by cellular guanylate kinase (GMK) and cellular nucleoside diphosphokinase to reach its triphosphate active form [7,8]. The triphosphate form of guanosine analogs are substrates for the viral DNA polymerase and to some extend to human DNA polymerases. Integration of antiviral nucleoside triphosphates through the DNA polymerase leads to chain termination or DNA misconformation [9]. DNA misconformation can lead to double strand breaks and cell death. The triphosphate form of guanosine analogs can also disturb DNA synthesis and repair in mitochondria, and can therefore also be toxic to non-dividing post-mitotic cells [10].

The concept to develop improved variants of HSV-TK has been investigated through the generation of TK variants by random mutagenesis targeting five key amino acid residues within the putative nucleoside binding site. Cellular expression of two HSV-TK variants, named 30-GMK and SR39, led to a markedly increased GCV sensitivity in vitro, and tumor cell killing in vivo [11,12].

In this study, we evaluated TK variants as safety switches for neural cell therapy. We also compared cell death induction by different antiviral nucleoside analogs. The efficacy of safety switches by these nucleoside analogs was then investigated in vivo by mouse brain transplantation of cells expressing the most efficient HSV-TK variant. Based on our results, we propose a high affinity, low toxicity safety switch technology.

## 2. Results

### 2.1. Generation of Cell Lines That Stably Express Thymidine Kinase

To reduce the safety risk of human embryonic stem cell (hESC) -based therapy, we are developing and optimizing genetically engineered systems that allow selective killing of proliferative cells, i.e., undifferentiated hESC or human neural progenitor cells (NPC). We first aimed to generate cell lines expressing different TK variants (Appendix A). Two recipient cell lines were used: (i) the human embryonic stem cell line HS420 (hESC), and (ii) ventral mesencephalon neural progenitor cell line: ReNcell [13]. To create stable transgenic cell lines, we generated lentiviral vectors that expressed the different TK variants C-terminally fused with the “sh ble” protein [14] (conferring zeocin resistance to transduced cells). TK variants were expressed under the control of either the ubiquitous ubiquitin promoter, or the cell cycle-dependent Ki67 promoter. 

### 2.2. Thymidine Kinase Variants and Ganciclovir-Induced Cells Death

Our first generation of suicide gene was engineered with the wild type thymidine kinase (wt-TK) [4]. To improve the suicide gene efficacy, we investigated different TK variants; wt-TK, 30-GMK, SR39, SR39h, and SR39h D116E variants. These sequences were chosen based on the analysis of the literature investigating HSV1-TK variants [15], as well as a gain of function mutation in the CMV UL-97 kinase [16]. The efficacy of each variants was determined as percentage of cell survival following prodrug treatment compared to non-treated hESC. For this purpose, we used the ATP-based assay, which measures cellular ATP content and thereby provides a reliable estimate of the number of healthy cells [17]. The 50% effective concentration (EC50) was defined as the drug concentration required to reduce the percentage of cell survival by 50%. Following 96h of exposure to GCV, the wt-TK killed proliferating hESC with an EC50 of 96 nM (Figure 1A,B). In order to optimize our system, we first replaced the wt-TK by the 30-GMK variant, which is a fusion between the 30 TK variant and guanylate kinase [15] (GMK). Note that GMK is mediating the second phosphorylation step of antiviral nucleoside analogs. However, by contrast to a previous study showing a higher efficacy to ablate proliferating rat C6 glioma cells [15], the 30-GMK variant was not superior to wt-TK in our experiments using hESC (EC50 110 nM). In contrast, the SR39 TK variant [11], which differs by only 5 amino acids from wt-TK (Appendix A), showed a higher efficacy to kill proliferating hESC, with an EC50 of 2 nM, i.e., a 50-fold reduction in EC50 compared to wt-TK. Microbial codon usage might not yield optimal results in human cells. We therefore adapted the SR39 sequence to a typical human codon usage. For this purpose, 218 of the total 376 codons of the SR39 sequence were changed (for details see Appendix A and Method section). We will refer to this sequence as the SR39h variant.

Surprisingly, SR39h was 10-fold more performant than SR39 with an EC50 of 0.2 nM (Figure 1A,B). In an attempt to further improve killing efficacy, we introduced a D116E mutation in the SR39h variant. This idea was based on the D605E mutation in the cytomegalovirus TK UL97, which increases the sensitivity of CMV to GCV [16]. This mutation is located in the substrate binding region, a region that share some similarity between UL97 from CMV and the TK from HSV-1. However, the D116E variant of SR39h demonstrated a slightly reduced efficacy as compared to the SR39h variant with an EC50 of 0.6 nM (Figure 1B). 

### 2.3. Characterization of the SR39h-Expressing HS420 Line

Given the high efficacy of SR39h/GCV mediated cytotoxicity compared to the other variants, we decided to further characterize the SR39h-expressing hESC. We first controlled the SR39h transgene insertion into the host cell DNA; PCR of genomic DNA clearly demonstrates a transgene insertion (Figure 2A). SR39h-expressing hESC colonies exhibited morphologic resemblance to those formed by the non-transduced hESC including tightly packed small cells, large nucleoli, and long-term growth potential (investigated for 10 passages). Note also that SR39h-expressing hESC could be cultured for at least 40 days as 3D neurospheres, differentiating towards a neuronal phenotype (data not shown). As expected for PSC, SR39h-expressing hESC and non-transduced hESC expressed pluripotency genes such as *OCT4*, *NANOG* and *SOX2* as shown by RT-PCR (Figure 2B). Upon differentiation towards a neuronal phenotype, these mRNAs strongly decreased. As expected, following neuronal differentiation leading towards a postmitotic phenotype, no Ki67 was observed in neurons. Note also the clear decrease of SR39h mRNA expression in the SR39h neurons, which is explained by the fact that SR39h expression is driven by the cell cycle-dependent Ki67 promoter. In line with the mRNA analysis, the hESC lines expressed the pluripotency markers at the protein level including OCT4, NANOG, and SOX2, as demonstrated by immunohistochemistry analysis (Figure 2C). 

Pluripotency markers’ expression points not only towards multilineage differentiation potential, but also towards a tumorigenic nature of the undifferentiated cells. To further investigate this question, we injected SR39h-expressing hESC cells into the striatum of NOD/SCID mice. Mice were sacrificed after 45 days and injection site within the brain was further analyzed. Visual inspection of the brains of injected mice showed clearly detectable tumor formation (Figure 2D, blue circle). Coronal sections after hematoxylin and eosin staining of brains showed tumors with a heterogeneous structure including cystic elements, compatible with teratoma formation (Figure 2E). Histopathological analysis and immunofluorescence staining of tumors (Figure 2F) revealed the presence of the 3 germ layers, detected by expression of the ectodermal marker α-βIII-Tubulin (βIII tub), the mesodermal marker α-smooth muscle actin (SMA) and the endodermal marker α-fetoprotein (AFP). Note that ectodermal markers were more strongly expressed as compared to endodermal and mesodermal markers, possibly due to the microenvironment of the transplantation site (brain).

### 2.4. Cell Cycle-Dependent vs. Ubiquitous Promoter: Impact on Cell Death Induction

To further study the expression of the suicide gene under the control of the cell cycle-dependent promoter, we used the ReNcell line. This is an immortalized human neural progenitor cell line that readily (i.e., within 21 days) differentiates into postmitotic neurons [13]. The reasons for the use of the ReNcell line for a part of our study are as follows: i. differentiation of hESC into post mitotic neurons is a long, complex, and costly process [18]; ii. exit from cell cycle takes at least 35–42 days for hESC (Appendix A), while this can be achieved within 21 days with ReNcells without the need of specific growth factors; iii. the use of ReNcells allows to study the activity of the suicide gene in a homogenous population of neural progenitor cells. As a first step, we investigated transgene expression under the control of the Ki67 promoter in ReNcell. Initially, we used eGFP, but the fluorescent GFP signal in differentiated ReNcell inactivated only very slowly and therefore did not allow to monitor promoter inactivation (data not shown). We therefore replaced eGFP by D2GFP, a fluorescent reporter with a short half-life [19]. As expected, compared to the non-transduced ReNcells (red), 92.4% of the Ki67-D2GFP-transduced undifferentiated ReNcells (violet) were GFP positive (Figure 3A–C).

As comparison, we also generated a construct where D2GFP was under the control of the ubiquitin promoter (Figure 3A). The flow cytometry analysis of undifferentiated ReNcells showed GFP intensities of 1.6 × 10^4^ and 1.5 × 10^5^ in undifferentiated Ki67-D2GFP and Ubi-D2GFP, respectively. Following 21 days of differentiation, more than 95% of the ReNcells transduced with Ki67-D2GFP were GFP negative (cyan), consistent with an efficient loss of Ki67 promoter activity upon ReNcells differentiation. The majority of the cells had a GFP intensity close to the background corresponding to the non-transduced cells. Nevertheless, we observed a small population with a slightly higher GFP intensity (second small peak, below the cut-off). This population had a forward scatter signal close to the undifferentiated cells and therefore might correspond to a population of less differentiated cells. In contrast, differentiated ReNcells transduced with Ubi-D2GFP (dark green) still expressed high levels of GFP with intensity comparable to undifferentiated cells (green) (Figure 3B,C). 

We then compared the ability of GCV to kill proliferating NPC that expressed the SR39h thymidine kinase under the control of the ubiquitin (ReNcells Ubi-SR39h) or the Ki67 promoter. (ReNcells Ki67-SR39h) (Figure 4). For this purpose, we first performed a dose response to GCV on undifferentiated NPC. After 96 h of GCV exposure, no difference in term of efficacy was observed between Ki67-SR39h- and Ubi-SR39-expressing cells, with an EC50 of ~2.1 nM. At high GCV concentrations (EC50: 12 µM), we also observed a killing of non-transduced proliferating ReNcells (Figure 4A). We then explored the kinetic of cell death with the suicide enzyme driven either by Ki67 or ubiquitin promoter for different concentrations of GCV. The rapidity of cell death induction but not its final extent was a function of the GCV concentrations. As shown in Figure 4B,C, there were no significant differences in term of cell death induction by GCV between the two constructs when 1 μM and 10 μM of GCV were administered. However, at highest concentration of GCV (100 μM), a slightly slower dynamic of cell death was observed in ReNcells Ki67-SR39h compared to ReNcells Ubi-SR39h. This was true for a short-term treatment at 24 h and 48 h, but not at the later time points at 72 h and 96 h (Figure 4D). Taken to together, our results confirm the ability of Ki67-SR39h to eliminate proliferative cells to a similar extend as SR39h driven by a ubiquitous promoter (Figure 4).

If the goal of suicide gene is to prevent tumor formation by eliminating proliferative cells, it should preserve grafted therapeutic cells, as well as the native healthy tissue.

We therefore investigated if a permanent expression of TK combined to GCV exposure can induce cells death in post-mitotic neurons. The SR39h enzyme was either under the control of the ubiquitin or Ki67 promoter. NPC were differentiated toward mature neural cells during 21 days to reach post-mitotic stage. Neurons were then exposed to increasing dose of GCV during 10 days. No cell death was observed in non-transduced neurons (wt) even at 100 μM, while cell death induction was detected from 10 μM in neurons expressing SR39h under the control of ubiquitin promoter (Figure 4E). In contrast, in neurons where SR39h is under the control of the Ki67 promoter, no cell death upon GCV exposure was observed. Upon 10 µM of GCV, Beta-III tubulin staining revealed drastic morphologic changes and damage in neurites of Ubi-SR39h neurons. In contrast, Ki67-SR39h neurons and wild type neurons were not sensitive to GCV and conserved a dense network of neurites (Figure 4F). These results demonstrate that in the presence of a TK, GCV damages post-mitotic neurons, albeit at higher concentrations than those needed to kill proliferating cells.

### 2.5. Optimization of Suicide-Inducing Nucleoside Analogs

GCV is a powerful and widely used suicide inducer in cells expressing different TK variants. However, a considerable toxicity limits its clinical usefulness [20,21]. We therefore investigated other nucleoside analogs requiring phosphorylation by HSV-TK for their conversion into active form: acyclovir (ACV), penciclovir (PCV), brivudine (BRVD), famciclovir (FAM), and valganciclovir (Val-GCV) [22,23]. The measurements in Figure 5 were done using an ATP-based assay (see above), which essentially measures the number of healthy cells.

Note, however, that ACV has a unique mechanism of action, being an absolute chain terminator due of the lack of the 3′-hydroxyl group. In contrast, the other compounds have a functional 3′ group (Figure 5A, green rectangles), allowing their incorporation into nascent DNA but leading to fatal modification of DNA structure [9]. We next investigated the effectiveness of the 6 different nucleosides analogs to induce cell death in undifferentiated hESC that express SR39h under the control of Ki67 promoter. FAM did not show any toxic effects (presumably because of a lack of metabolism in our cellular system). ACV and BRVD killed SR39h-expressing cells only at high concentrations, 2.9 µM and ~60 µM, respectively (Figure 5B). Val-GCV, PCV, and GCV killed SR39h-expressing cells with EC50 values 32 nM, 2.4 nM, and 0.2 nM, respectively (Figure 5B). EC50 values are summarized in panel 5C.

We next investigated the impact of ACV, PCV, and GCV on non-transduced hESC and neuroprogenitors (proliferating ReNcells). Clearly, there was a GCV-mediated cell toxicity in the low micromolar range. In contrast, we did not observe toxicity with ACV and PCV up to 100 µM (Figure 6A). To further understand the impact of nucleoside analogs on non-transduced cells, we performed γ-H2A.X immunostaining (Figure 6B). There was a graded response, virtually identical in hESC and in neuroprogenitors. With 48 h of nucleoside (100 µM) exposure, we observed a strong γ-H2A.X signal located in the nucleus in response to GCV, but only weak signal in response to PCV and ACV (Figure 6B). In addition, we investigated the morphology and size of non-transduced hESC colonies upon nucleoside treatment. We did not observe an effect of ACV and PCV, however, upon GCV treatment, we observed smaller colonies (Figure 6B) and numerous detached cells (not shown). Taken together, our findings confirm considerable toxicity of GCV, even in the absence of viral TK. Note that this was not the case for PCV despite the fact that it is a potent inducer of cell death in thymidine kinase-expressing cells. 

### 2.6. In Vivo Validation

Given the encouraging in vitro results, we next investigated whether this newly developed safety switch system (SR39h and different nucleoside analogs) could prevent tumor formation in vivo. For this purpose, 20,000 undifferentiated Ki67-SR39h-expressing hESC were transplanted into the striatum of NOD/SCID mice. Two days after transplantation, mice received guanosine analogs (ACV, PCV and GCV) treatment by intraperitoneal injection: 40 mg/kg/day, five consecutive days a week during a two-week period (Figure 7A). A group of control mice received only a NaCl solution. Note that no significant differences in weight among the different groups was observed following guanosine analogs treatment (Appendix A). Forty-seven days post-transplantation, as expected (see Figure 2D,E), massive tumors were observed in control mice group (Figure 7B). Mice treated with ACV developed large tumors, similar to those seen in the control group. Human cytoplasmic marker (HCM) staining confirmed the human (i.e., hESC-derived) origin of these tumors. Tumors cells were predominantly Ki67 positive with a rosette organization as frequently found in tumors of the central nervous system. As opposed to control and ACV-treated mice, no major tumors formed by grafted hESC were observed in PCV- and GCV-treated mice, and only few HCM- and Ki67-positive cells were observed at the site of injection (Figure 7C). More specifically, we did not observe any structures resembling tumors in GCV treated cells; however, we observed a small nucleus-rich region in one of the PCV-treated mice (Appendix A, small purple region indicated by the red arrow).

## 3. Discussion

Pluripotent stem cells (PSC) hold great promise for future regenerative therapies, in particular in the context of neurodegenerative diseases. However, their proliferative potential is also a potential hazard, as the risk of tumor formation must be controlled before a transfer into clinics. We have previously proposed a safety switch technology based on the thymidine kinase (TK) form Herpes Simplex virus type 1 (HSV-1) and the nucleoside analog ganciclovir (GCV) as a suicide inducer. This first-generation safety switch technology works well and allows reliable prevention of tumor formation in mice transplanted with embryonic stem cells (hESC). However, the first-generation safety switch has two limitations: (i) with the wild type TK, the concentrations of nucleoside analogs required to induce cellular suicide are relatively high, (ii) GCV is a relatively toxic compound, both in vitro [25] and in vivo [26] and replacement by a less toxic nucleoside analog is desirable. As a result of this study, we propose a second-generation safety switch, which overcomes both of these problems: (i) to induce cell death in SR39h-expressing cells, much lower concentrations of nucleoside analogs are needed, and ii) penciclovir (PCV) is a safe and relatively non-toxic compound, comparable to acyclovir (ACV) [27]. 

Previous studies have demonstrated that the activity of HSV-1 TK as a suicide gene can be markedly improved by introducing molecular changes [11]. One strategy used was the use of a fusion protein consisting of TK and guanylate kinase (GMK) sequence, facilitating the second phosphorylation step following the initial TK phosphorylation. In our study, this strategy did not yield improved results, possibly due to the fact that the enhancement of the TK suicide gene activity through the GMK fusion proteins depends on the levels of endogenous GMK in a given cell type. In contrast, the SR39 mutations in the phospho-transfer regions of TK (amino acids 159–169 [11]) led to a massive increase in suicide gene activity in our experiments. As viral RNA sequences are potential targets of cellular host defense mechanisms, we proceeded to change the cDNA sequence without changing the amino acid sequence to a pattern more typical for human codon usage (SR39h). This mRNA sequence led to a marked increase in suicide gene activity, possibly due to either increased RNA levels, or to increased transduction.

Note that, in this study, TK variants were inserted by lentiviral vectors followed by polyclonal selection. Thus, there are multiple insertion sites, and effects observed in this study are unlikely to be due to an effect of a specific transgene insertion.

In our study, we investigated six nucleoside analogs for the capacity to induce cell death in SR39h-expressing cells. ACV, and BRVD induced only very little cytotoxicity in TK-expressing cells, despite showing an anti-Herpes activity in infected cells [28,29]. This might be due to a good selectivity for viral over mammalian DNA polymerases or—in the case of ACV—due to its mechanism of action as a chain terminator (see also below). FAM and Val-GCV are pro-prodrugs designed for oral uptake that require metabolism by hepatic and/or cellular enzymes for liberation of PCV or GCV, respectively. FAM did not show any activity, presumably because of the necessity of hepatic metabolism for it activation [30]. Val-GCV had a relatively weak, but clearly discernible activity. Finally, only PCV and GCV were strong suicide inducers. 

To explain why PCV and GCV are efficient suicide genes, while ACV is not, we consider several, mutually not exclusive, possibilities: (i).Differences in mode of action: ACV is thought to stop DNA synthesis through chain termination, but there is no wide-spread activation of cell death mechanism. More specifically, no fragmentation of host cell DNA in response to ACV was observed [31]. In contrast, PCV and GCV, which allow continuation of elongation despite integration of “wrong nucleosides” have the potential to lead to multiple double strand breaks within the nascent DNA. Accordingly, both compounds have been shown to induce fragmentation of host cell DNA [31]. These differences in mode of action are explained by the presence in PCV and GCV of a 3′-hydroxyl group, which is absent in ACV. The 3′-hydroxyl group, which is also present in naturally occurring nucleosides, is necessary for the continuation of DNA elongation. Thus, elongation with “wrong nucleotides” rather than chain termination might underlie the induction of cell death.(ii).Differences in TK activity on different nucleosides: Wild type TK has indeed a much lower affinity for ACV than for GCV; however, this is not the case for SR39 [12]. Thus, the low capacity of ACV to induced cell death in SR39-expressing cells is not due to a decrease TK phosphorylation of ACV. (iii).Differences in selectivity for viral vs. mammalian DNA polymerases: A lower affinity for the mammalian DNA polymerase could account for the fact that ACV does not induce cell death. From a theoretical point of view, this is a pertinent explanation, but there are no systematic studies addressing this question. One article compares affinities of nucleosides for viral and mammalian DNA polymerases, based on numbers from different publications [32]. In this context, it is important to point out that humans have 16 different DNA polymerases for replication and repair of nuclear DNA, as well as one additional DNA polymerase for mitochondrial DNA replication [33]. At this point, it is not clear which of these human DNA polymerases are able to use GCV triphosphate and PCV triphosphate as substrate for DNA synthesis. Given the rapid integration of GCV into the DNA dividing cells, it is tempting to speculate that DNA polymerase ε and δ might be involved. However, given the cell death induction by GCV in post-mitotic SR39h-expressing neurons, it appears that there is also a role for other polymerases, either those involved in DNA repair in post-mitotic cells [34] or the mitochondrial DNA polymerase. However, the markedly higher GCV concentrations were necessary to kill SR39h-expressing post-mitotic neurons. There are several explanations for this difference in terms of EC50, for example a different affinity of different polymerases for GCV triphosphate, or the fact that DNA repair in post-mitotic cells requires much less polymerase activity than DNA synthesis during replication in dividing cells. However, we cannot exclude a different mode of action of GCV in post-mitotic cells.

It was also reported in the literature that permanent expression of viral TK can disrupt cell through alteration of dNTP pool homeostasis leading to cell death [35]. We did not observe such issue with both NPC and hESC expressing the TK either under the control of ubiquitin or Ki67 promoter. A possible explanation is the fact that we used low MOI < 0.5 when generating our cell lines to minimize the risk of multiple insertions. Moreover, we used the SR39h mutant, which is present five amino acid changes in the substrate binding region. It is possible that these mutations enhance the GCV affinity at the expense of thymidine monophosphate (dTMP), thymidine diphosphate (dTDP) as well as thymidine triphosphate (dTTP), thus reducing the risk of unbalanced dTNP pool.

Thus, with respect to death induction in TK-expressing cells, PCV behaves similar to GCV. However, in other aspects, PCV resembles more to ACV than to GCV. ACV and PCV are antivirals with a relatively narrow spectrum, essentially limited to α-herpesviruses [23], while GCV is also active on β-herpesviruses, in particular CMV [36]. This difference between PCV and GCV with respect to toxicity in TK-negative cells has been observed in vitro (Figure 6A,B) and in vivo [37]. According to one study investigating this question, GCV, but not PCV, is a potent inducer of sister chromatid exchange, already at concentrations far below those impairing the proliferative activity [25]. This suggests that even a very limited phosphorylation of GCV by a mammalian kinase could be sufficient to induce toxicity. 

Both PCV and GCV showed impressive anti-tumor activities in vivo (Figure 7C). However, in one out of five PCV treated mice (Appendix A, small purple region indicated by the red arrow), we observed a small nuclei-rich region, suggesting the presence of proliferating cells. Thus, it is possible that PCV (at equivalent concentrations) is slightly less potent as a suicide inducer, as compared to GCV. This might reflect the slightly higher EC50 values for PCV observed in the in vitro killing curves (Figure 5B,C) and slightly higher PCV concentrations might be needed for a complete killing of all proliferating cells. We think, however, that this possible disadvantage of PCV is largely outweighed by its advantage, namely the low toxicity. Thus, PCV treatment of patients will be possible with higher doses as well as for longer treatment periods without the risk of toxicity, which will be a real issue with GCV. Indeed, daily i.v. doses for ACV (a compound with a tolerability similar to PCV) can go up to 45 mg/kg/day, without major side effects. This is in contrast with GCV, for which at the maximal recommended doses (10 mg/kg/day), there are already considerable side effects. Thus, we believe that PCV will ultimately be the optimal compound to be used in our safety switch system. 

In summary, we have developed a second-generation safety switch technology, which uses a suicide gene with a very high affinity for nucleoside analogs to act as a suicide inducer for neural cell therapy. This technology provides a good therapeutic window and is expected to have few side effects. We therefore think that it is ready to rapidly advance towards clinical studies. 

## 4. Materials and Methods

### 4.1. Human Embryonic Stem Cells (hESChESC): Maintenance Culture and Neural Differentiation

HS420 hESC (used from passage 34 to 40, kindly provided by Outi Hovatta, Karolinska Institute, Stockholm, Sweden) was expanded on recombinant Laminin 521 (rh LAM521, 0.5 µg/cm^2^ and working concentration 2.5 mg/mL), (ThermoFisher, Geneva, Switzerland) in Stemflex, a feeder-free culture medium (Invitrogen, Basel, Switzerland). Medium was changed every 2–3 days to maintain pluripotency. Cells were passaged with enzymatic procedure Accutase (Invitrogen, Basel, Switzerland) and replated with 10 µM Rho-associated protein kinase inhibitor (ROCK inhibitor) (Y-27632; Axon Medchem, Groningen, Netherlands) during 24 h before removal. The hESC line were differentiated toward neurons during five weeks to reach post mitotic stage (for details, see our previous paper [18]).

### 4.2. Culture of ReNcells

The ReNcell line (ReNcell^®^ VM, Chemicon, Merckmillipore, Darmstadt, Germany) was derived from 10-week-old gestational ventral mesencephalon brain tissue and immortalized with the v-myc oncogene. ReNcells were routinely expanded for experimental work on T75 cm^2^ coated with Matrigel 1/100 (Corning, Lonay, Switzerland) and cultured with Neurobasal media (Gibco, ThermoFisher, Geneva, Switzerland), supplemented with B2 medium (Gibco, ThermoFisher, Geneva, Switzerland), 2 mM L-Glutamine, 1 mM Sodium pyruvate, 1× Non-Essential Amino Acids Solution and 1% penicillin-streptomycin antibiotics (all from Gibco). To maintain ReNcells at neuro-progenitor stage, additional growth factors were added to Neurobasal media: Fibroblast Growth Factor-basic (bFGF) (20 ng/mL, Invitrogen, Basel, Switzerland) and Epidermal Growth Factor (EGF) (20 ng/mL, Sigma, Merk, Eysins, Switzerland). ReNcells were differentiated by removing the growth factor b-FGF and EGF. ReNcells differentiated into neurons and reach post mitotic stage in 6 to 8 days in Neurobasal media supplemented with B27 medium, 2 mM L-Glutamine, 1 mM Sodium pyruvate, 1× Non-Essential Amino Acids Solution and 1% penicillin-streptomycin antibiotics (all from Gibco, ThermoFisher, Geneva, Switzerland).

### 4.3. Humanization of TK DNA Sequence

Viral-TK DNA sequence was humanized using GenScript web-based algorithms (GenSmart™ Codon Optimization). More than 200 factors involved in gene expression, including GC content, codon usage and content index, RNase splicing sites, and cis-acting mRNA destabilizing motifs were screened and used for optimal humanized cDNA design. 

### 4.4. Lentiviral Vector Construction

The lentiviral plasmid was generated by recombinational cloning of the destination vector pCLX-DEST, from a pENTR-promoter and a pENTR-Gene Of Interest using the LR Clonase II (Invitrogen, Basel, Switzerland). The pENTR promoter contains either Ki67, a cell cycle promoter or Ubiquitin promoter (pENTR-Ki67/Ubi). The pENTR-Gene Of Interest contains either D2GFP or the fusion gene Tkvar-Sh (pENTR-D2GFP/Tkvar-Sh). Tkvar-Sh refers to a Herpes simplex thymidine kinase variant fused with the “Sh ble” gene conferring zeocin resistance.

Lentiviral particles were generated by co-transfection of pCLX-Ki67/Ubi-TKvar-Sh, the packaging plasmid: psPAX2 encoding gag/pol and the envelope plasmid: pCAG-VSVG on HEK 293T cells. HEK 293T cell line was cultured in high-glucose Dulbecco’s modified eagle medium supplemented with 10% fetal calf serum, and 1% L-glutamine (all from Gibco, ThermoFisher, Geneva, Switzerland). The next day following the transfection, fresh media without fetal calf serum was added, then 24 h later, the media containing lentiviral vectors were harvested, centrifugated at 2400 rpm during 10 min, filtered with 0.45 µm filter, and stored at −80 °C until use.

### 4.5. Cell Transduction and Selection

ReNcells and hESC lines were transduced during 6h with lentiviral vectors at a MOI (multiplicity of infection) below 0.5. Cell lines expressing TK variants were selected using zeocin. Zeocin exposure was maintained during 10 days, and concentrations were 3.5 µg/mL for HS420 and 125 µg/mL for ReNcells. Optimal zeocin concentrations were obtained by performing dose killing curves in wild-type HS420 and ReNcells. Genomic insertion of the TK transgenes was verified by PCR (data not shown). Cell lines expressing D2GFP were selected by cell sorting using MoFlo Astrios (Beckman Coulter, Brea, CA, USA). There was no impact of transgene expression on cell differentiation (Appendix A).

### 4.6. ATP-Based Cytotoxicity and Kinetic of Cell Death Assay 

Cell culture plate (96 wells plate Corning #3907, Corning, Lonay, Switzerland) were pre-coated with 60 μL per well of Poly-L-ornithine solution (Sigma, Merk, Eysins, Switzerland) at 37 °C for 30 min. Plates were washed twice with DMEM-F12 (Gibco, ThermoFisher; Geneva, Switzerland), then coated with Matrigel (Corning, Lonay, Switzerland) 1/100 in DMEM-F12 and incubate 2 h at 37°. Cells were detached using Accutase and seeded at 2000 cells/well in serum-free culture medium containing 10 µM of ROCK inhibitor. At 24 h, the cell culture was continued without ROCK inhibitor. To determine the EC50 of antiviral nucleoside analogs, 24 dilutions were tested. Drugs were prepared by serial dilution (two-fold) in 100% DMSO. At 48 h after seeding, cells were exposed to antiviral drugs for 96 h. For EC50 assays, the medium was changed daily for 4 days, then cell viability was assessed using an ATP-based assay. EC50 values for each construct were calculated using a four parameters equation (GraphPad, Prism8, San Diego, CA, USA). The same procedure was used for the cell death kinetic determinations except the treatment duration. Luminescence was measured using Spectra L384/L96 (0.2 sec exposure, Molecular devices, Biberach an der Riß, Germany).

### 4.7. RNA Extraction and Quantitative Real Time Polymerase Chain Reaction

Total RNA was extracted using the RNeasy MiniKit QIAGEN according to the manufacturer’s protocol (Invitrogen, Basel, Switzerland). The cDNA were produced by reverse transcribed 500 ng of extracted RNA using PrimeScript RT reagent Kit (Takara Bio USA, Mountain View, CA, USA). PCR reaction reactions were performed in a Biometra thermocycler (Analytik Jena, Göttingen, Germany), with allTaq polymerase mix (Quiagen, Hombrechtikon, Switzerland), 250 nM primers and 1 μL of cDNA. The primer sequences used for qRT-PCR are listed in Table 1 below.

### 4.8. Immunocytochemistry

Briefly, HS420 hESC and ReNcells cultured on cover slips coated with Matrigel (1/30 in DMEM) were fixed in 4% paraformaldehyde in phosphate buffered saline (PBS) for 10 min at room temperature, washed and processed for conventional immunocytochemistry. Cells were incubated with a primary antibody Oct4 (rabbit polyclonal 1/200, Abcam, ab18976), Nanog (mouse monoclonal, 1/800, Abcam ab 171380), Sox2 (goat polyclonal 1/400, ThermoFisher, tf-pa5-18406,), Beta-III tubulin (mouse monoclonal, 1/1000, Sigma, T8660), γ-H2A.X (rabbit polyclonal 1/1000, Abcam ab11174), Ki67 (rabbit monoclonal 1/250 Abcam, ab16667) diluted in PBS-Triton 0.1% for 1 h at room temperature (RT). Detection of primary antibodies was performed using appropriate species-specific Alexa 488-, Alexa 555- or Alexa 647-labeled secondary antibodies for 1 h at RT in the dark. The cells were finally stained with Hoechst 33342 or 4′-6-diamidino-2-phenylindole (DAPI) for nuclei identification. Slides were mounted in Fluorosave (Calbiochem, Merck Millipore). Controls included examination of the cell or tissue autofluorescence and omission of the first antibody. Images of the immunostained cells were captured on a Zeiss Axioskop 2 fluorescence microscope with an Axiocam Color HRC detector (Zeiss, Feldbach, Switzerland).

### 4.9. Flow Cytometry

ReNcells were cultured as described before, then detached by enzymatic procedure with TrypLE. Cells were collected by centrifugation at 1000 rpm for 5 min and resuspended in phenol red free DMEM/F12. The parameters for gating cell populations by flow cytometry were established using non-transduced ReNcells cells. D2GFP fluorescence was measured in single-cell suspensions of ReNcells on Gallios Flow Cytometer (Beckman Coulter), (FL1, 488–561 nm) and data analysis by Flowjo software.

### 4.10. Stereotaxic Engraftment and Treatment with Antiviral Nucleoside Analogs

All experiments conducted on mice were compliant to local ethical committee guidelines. NOD-SCID mice (Charles River, Écully, France) were anesthetized with Isoflurane and treated with Lidocaine, a local anesthetic.

Single cell transplant: Undifferentiated HS420 (expressing TK variants) were detached by TrypLE and concentrated at 20,000 cells/μL in Stemflex. 20,000 cells (1 μL) were injected into the right striatum using a 26-G Hamilton syringe (Hamilton, BGB Analytik AG, Boeckten, Switzerland). Injection coordinates for striatum were: bregma = 0.6 mm, mediolateral = 2 mm, dorsoventral = 3 mm. 

Two days post stereotaxic injection mice were treated by intraperitoneal injection of ACV (Labatec, Meyrin, Switzerland), PCV (Enamine, Kyiv, Ukraine), or GCV (Cymevene, Roche, Bâle, Switzerland): 40 mg/kg (five consecutive days a week during two weeks) using Insulin micro-syringe 30G (Becton Dickinson, Allschwil, Switzerland). 

### 4.11. Mouse Sacrifice and Brain Analysis

Thirty days after termination of treatment with the respective antiviral nucleoside analogs or control (NaCl), mice were sacrificed by intracardiac perfusion of paraformaldehyde 4% in PBS. Brains were removed from the skull and post-fixed in the same fixative overnight.

Paraffin embedded brains were sectioned (coronal sections 7 µm thick) and processed for hematoxylin and eosin or cresyl violet staining stain aiming morphological assessment of the graft. Sections were mounted in Eukitt (Kindler GmbH, Germany) and observed under microscope (see below)

For immunohistochemistry, slides were first deparaffinized with Xylene (Reactolab, Servion, Switzerland) and ethanol bath, 5 min each. Antigen retrieval was achieved with citrate method; 3 cycles in the microwave at 620 watts during 5 min (citrate buffer pH: 6, 0.01 M). 

The primary antibody diluted in PBS with 0.3% Triton was incubated at 4 °C overnight in agitation. The following primary antibodies were used: Ki67 (monoclonal rabbit 1/100, Abcam, ref ab 16667) and α-HCM (stem 121 1/200, Takara, Saint-Germain-en-Laye, France), α-fetoprotein AFP (1/50, Santa Cruz, sc-8399), α-Smooth Muscle SMA (1/250, Sigma, A2547), α-Beta-III tubulin (1/1000, Sigma, T8660). Detection was performed using Alexa 488 or 555-labeled secondary antibody (room temperature for 1 h). Controls included examination of the tissue autofluorescence and omission of the first antibody. Cell nuclei were stained with DAPI. Sections were mounted in Fluorosave (Calbiochem, VWR, Nyon, Switzerland) and observed with an Axioscop 2 plus microscope equipped with appropriate filters, Axiocam color camera, and Axiovision software (Zeiss, Feldbach, Switzerland).

## Figures and Tables

**Figure 1 cells-11-00502-f001:**
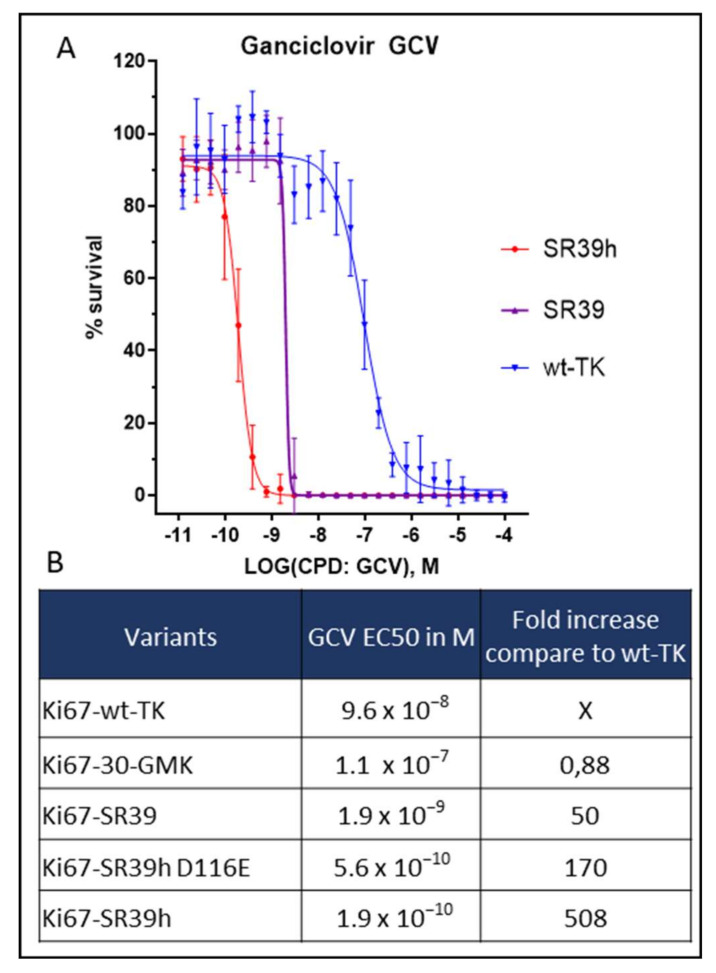
EC50 of ganciclovir (GCV) in cell expressing different thymidine kinase (TK) variants. Five different variants of TK from Herpes Simplex virus were investigated for their efficacy as a suicide gene in embryonic stem cells HS420 (hESC), in the presence of GCV: wt TK, 30 GMK, SR39, SR39h D116E, SR39h. All TK variants were expressed under the control of the cell cycle-dependent Ki67 promoter. (**A**) GCV EC50 curves in hESC expressing TK, SR39, or SR39h (*n* = 3; GCV exposure time was 96 h). Number of surviving cells was monitored using bioluminescent ATP-based assay. (**B**) Summary table of EC50 values of all tested TK variants.

**Figure 2 cells-11-00502-f002:**
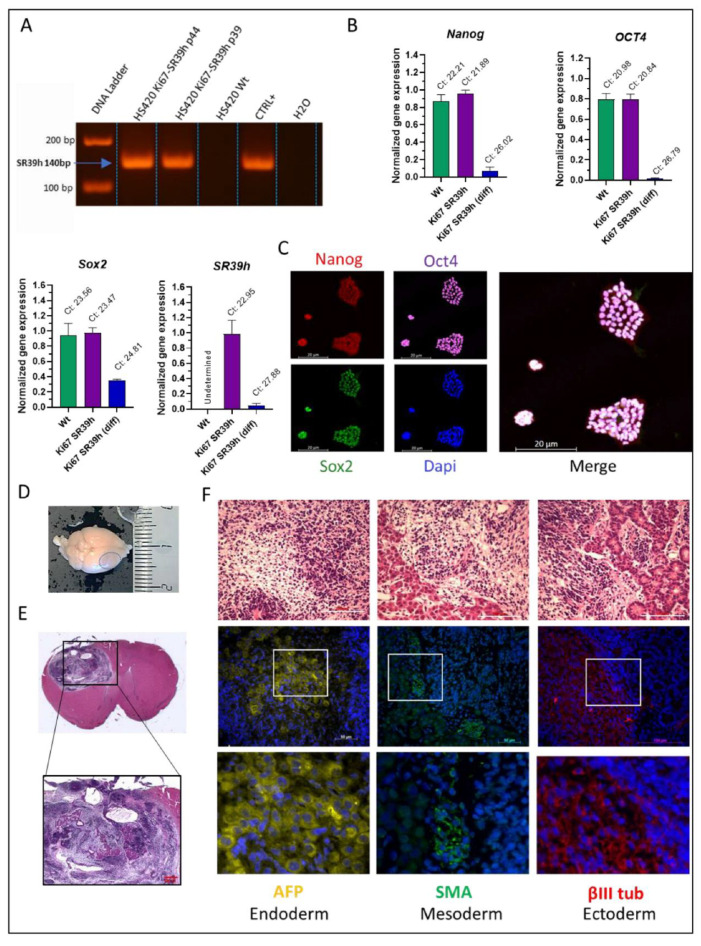
Characterization of the SR39h-expressing hESC line. (**A**) PCR analysis of SR39h transgene insertion in genomic DNA. (**B**) RT-PCR analysis of the expression of pluripotency genes (*n* = 3), *OCT-4, NANOG, SOX2*, and SR39h in SR39h-expressing hESC (undifferentiated and differentiated), as well as control wild type cells (undifferentiated). (**C**) Immunofluorescence staining of pluripotency markers (OCT4, SOX2, NANOG), nuclei were counterstained with Hoechst. The merge panel consists of all four staining. Scale bar 20 µM. (**D**) Macroscopically visible teratoma (blue circle) after brain extraction from the skull bone. (**E**) Hematoxylin and eosin staining of coronal slice of mice brain. (**F**) Upper panel; coronal sections brain with teratoma stained with hematoxylin and eosin. Scale bar 100 µM. Lower panels; immunostaining: α-fetoprotein (AFP) for endoderm-derived tissue, α-smooth muscle actin (SMA) for mesoderm-derived tissue and Beta-III tubulin (βIII tub) for ectoderm-derived tissue.

**Figure 3 cells-11-00502-f003:**
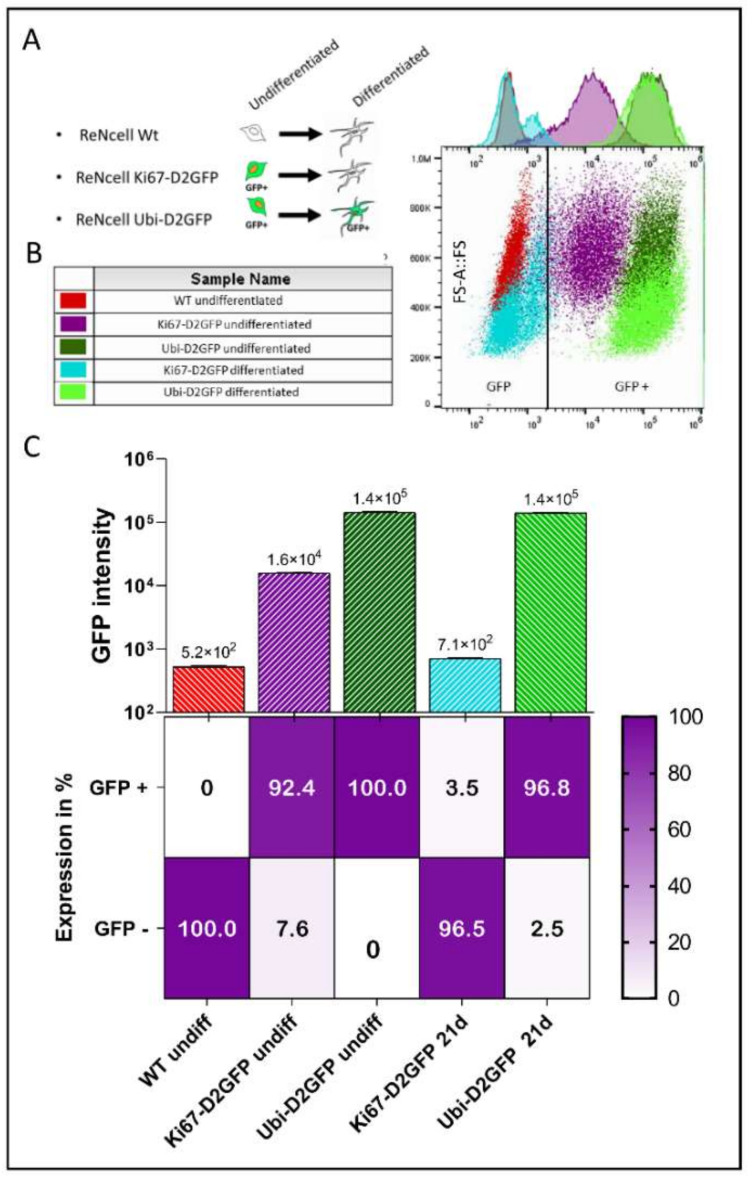
Reporter gene expression: Ki67 versus ubiquitin promoter. D2GFP was expressed in ReNcells (immortalized neural precursor cells) under the control of the two different promoters. Wild type ReNcells were used as control. (**A**) Schematic representation of the expected reporter gene (D2GFP) expression driven by the cell cycle-dependent Ki67 or the ubiquitous ubiquitin promoters, (**B**) Scatter plots and representative histograms from FACS showing D2GFP intensity on the x-axis, and forward scatter (FS) on the y-axis. Data shown are representative of three independent experiments. (**C**) Average of GFP intensity measured by FACS analysis (*n* = 3); all data are represented as the mean ± SD, and corresponding percentage of GFP positive and negative cells.

**Figure 4 cells-11-00502-f004:**
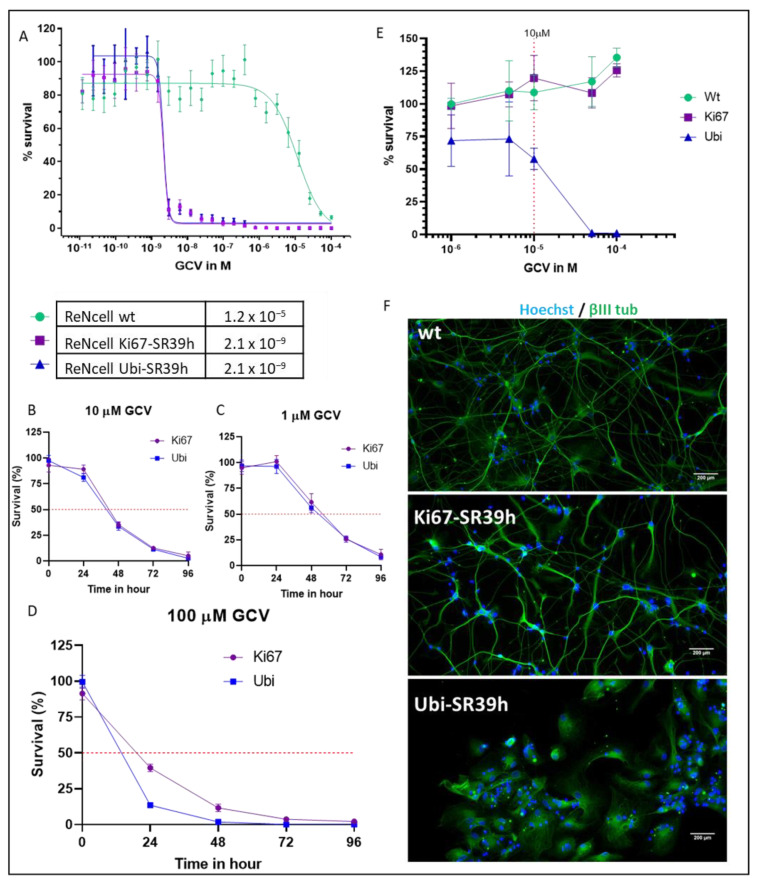
EC50 and kinetics of GCV-induced cell death in proliferating and post-mitotic ReNcells. SR39h was expressed in ReNcells (immortalized neural precursor cells that can be differentiated to post-mitotic neurons) under the control of the two different promoters: Ki67 or ubiquitin. Wild type ReNcells were used as control. (**A**) Dose response of GCV in proliferating neural precursor cells after 96 h of drug exposure, EC50 were expressed in M. Bioluminescent ATP-based assays were performed, and 24-point dose response curves were generated (*n* = 3). (**B**–**D**) Kinetics of cell death of SR39-expressing ReNcells cells either under the control of cycle promoter Ki67 or ubiquitin promoter with increasing GCV concentrations of 1, 10, and 100 µM. Cell viability was measured every 24 h using a bioluminescent ATP-based assay (*n* = 3). (**E**) ReNcells were differentiated toward post-mitotic neurons for 21 days then treated with GCV for 10 days. Cell viability was measured using an ATP-based assay. Results are mean + SD of three independent experiments performed in triplicates. (**F**) Immunostaining, Hoechst in blue and Beta III tubulin (βIII tub) in green, ReNcells were differentiated toward neurons for 21 days, then treated with 10 µM of GCV during 10 days.

**Figure 5 cells-11-00502-f005:**
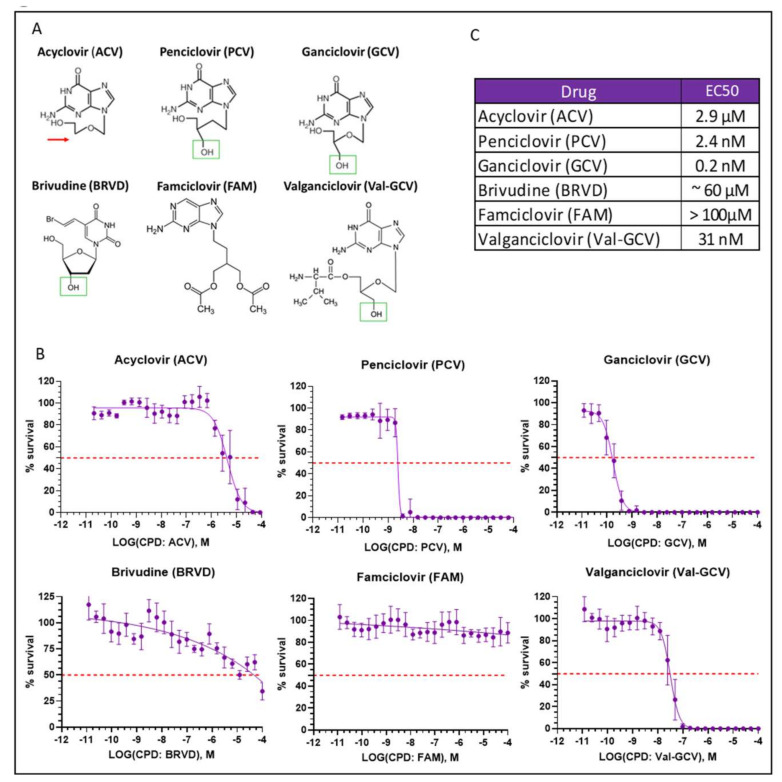
Cell death induction by antiviral nucleoside analogs in SR39h-expressing hESC. (**A**) Chemical structures of the antiviral nucleosides acyclovir (ACV), penciclovir (PCV), ganciclovir (GCV), brivudine (BVDR), famciclovir (FAM), and valganciclovir (Val-GCV). The green boxes highlight 3′-hydroxyl groups (required for chain elongation). ACV does not have a 3′-hydroxyl group (red arrow). (**B**) Nucleoside-induced cell death in SR39h-expressing cells. Cells were exposed to increasing concentrations of the respective nucleosides for 96 h and cell viability was assessed using an ATP-based assay. Results are mean + SD of three independent experiments performed in triplicates. The dotted line indicates 50% cell survival. (**C**) Summary table of the respective EC50 values.

**Figure 6 cells-11-00502-f006:**
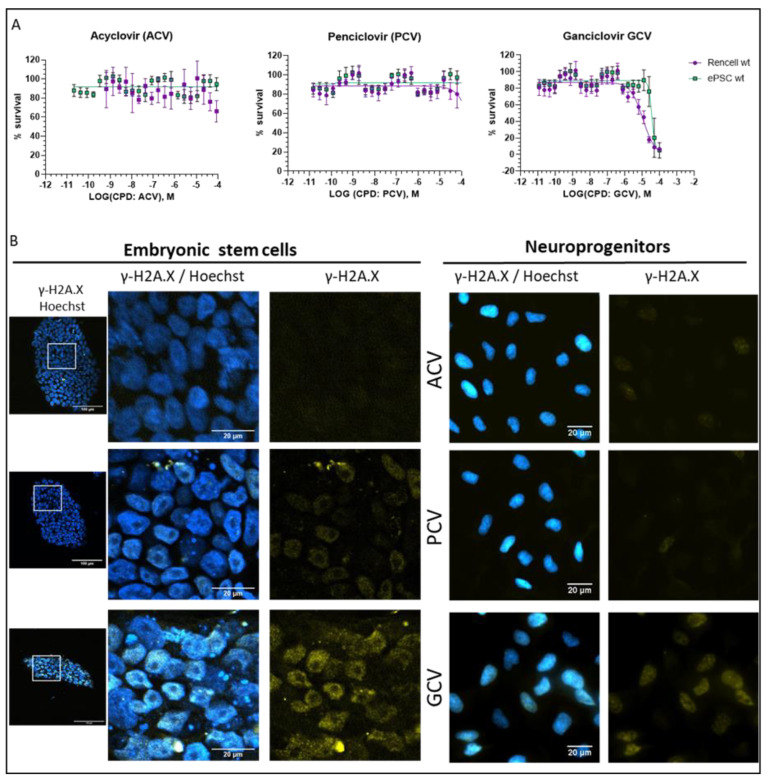
Impact of antiviral nucleoside analogs on non-transduced hESC and neuroprogenitor cells (**A**) Toxicity of ACV, PCV, and GCV in non-transduced hESC and neuroprogenitors (proliferating ReNcells), detected by an ATP-based assay (nucleoside exposure time 96 h, *n* = 3). (**B**) DNA damage induced by guanosine analogs ACV, PCV and GCV in non-transduced cells. Cells were exposed to 100 µM of the respective nucleoside for 48 h. DNA damage was monitored using γ-H2A.X marker [24] in yellow and nuclei were counterstained with Hoechst. The white box in the first column indicates the area of the zoom for the hESC colonies.

**Figure 7 cells-11-00502-f007:**
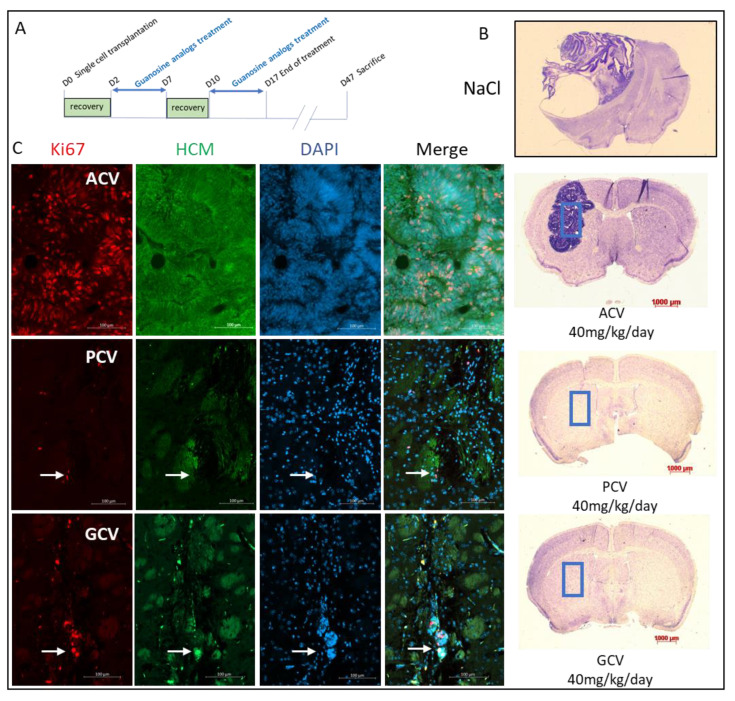
in vivo validation of suicide gene and inducers. (**A**) Schematic diagram of the experimental setup to detect tumor elimination in vivo. NOD/SCID mice were transplanted with 20,000 SR39h-expressing hESC through stereotactic injection into the striatum. 48 h post transplantation, mice were intraperitoneally injected with the respective nucleoside (40 mg/kg) for five consecutive days a week during a two-week period. Control mice were treated with a NaCl solution. Mice were sacrificed 47 days post transplantation. (**B**) Cresyl violet staining of a brain section. Tumor formation is clearly visible in control and ACV-treated mice (right hemisphere, where human hESC had been injected). (**C**) Immunostaining of brain coronal section (sections from ACV-, PCV-, and GCV-treated mice are shown in the upper, middle, and lower row, respectively). Staining with the proliferation marker Ki67 and with the human cytoplasmic marker (HCM) are shown in red and green, respectively. Nuclei were counterstained with DAPI (blue). White arrows show residual proliferative cells form the graft. A merge of the three staining is shown in the right column. Each experimental group consisted of five mice.

**Table 1 cells-11-00502-t001:** Primer used for qRT-PCR.

	Forward	Reverse
*GAPDH*	CAAGATCATCAGCAATGCCT	CTTCCACGATACCAAAGTTGTC
*SR39h*	CAGCGAGACAATCGCCAAC	CCAGCACAGCATCTGTCAC
*OCT4,*	CTTGCTGCAGAAGTGGGTGGAGGAA	CTGCAGTGTGGGTTTCGGGCA
*NANOG*	CAAAGGCAAACAACCCACTT	TCTGCTGGAGGCTGAGGTAT
*SOX2*	GCCGAGTGGAAACTTTTGTCG	GGCAGCGTGTACTTATCCTTCT

## Data Availability

All data generated or analyzed in this study are included in the manuscript and Appendix A.

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
