# Peer review of "Optimization of Thymidine Kinase-Based Safety Switch for Neural Cell Therapy"

_cells, 2022, doi:10.3390/cells11030502_

Round 1

Reviewer 1 Report

In this work Locatelli and colleagues developed a mechanism for controlling the proliferation of ePSCs.  After my last revision, the manuscript was improved.  There are a couple of issues I believe need further clarification:

Major:

1- The insertion of the plasmid containing Ki67-SR39h in the cells genome was not demonstrated in your work.  I believe this is especially important, since this plasmid differs from the one you used in your previous work.  I believe RT-PCR is not enough and does not account for the insertion/plasmid location inside the cell.  Please consider DNA-sequencing to prove this.

Results

4- Please insert the number of independent experiments performed for the results presented in figures 2A.

Discussion

8- Regarding the in vivo assay, was animal behavior monitored throughout the experiment?  Can these results be provided?

Author Response

"Major: 1- The insertion of the plasmid containing Ki67-SR39h in the cells genome was not demonstrated in your work. "

As already pointed out in our previous response, we did not work with a clone, but with cells polyclonally transduced by a lentivector. Thus, we could show insertion into genomic DNA (Fig. 2A), but as the site of insertion differs from one cell to the other, there is not a single site of insertion. 

"Please insert the number of independent experiments performed for the results presented in figures 2A."

Results shown in Fig.2A were repeated twice (once at passage 39 and once at passage 44) confirming the stable integration.

"Regarding the in vivo assay, was animal behavior monitored throughout the experiment?  Can these results be provided?"

Data on animal weight are shown in supplementary figure 3. We have not observed any abnormal behavior of the transplanted animals, however quantitative testing was not performed.

Reviewer 2 Report

In this version the authors made some corrections and revised the language and writing, however, I did not find the data to answer my questions. Therefore, my concerns about in vivo transplantation and cell lines/types are still there. Particularly, I am not convinced that whether the system could indeed kill the proliferating cells (the tumor-generating cells) while keep the post-mitotic functional neurons alive.

By the way, the “misnomer” “embryonic stem cell” has been used in 10,000 publications while “embryonic pluripotent stem cell” is just mentioned in very few papers (from PubMed).  

Author Response

"I am not convinced that whether the system could indeed kill the proliferating cells (the tumor-generating cells) while keep the post-mitotic functional neurons alive."

We have demonstrated in vitro (Fig. 4) and in vivo (Tieng et al, 2016 doi:10.1038/mtm.2016.69) that thymidine kinase under the control of a cell cycle-specific promoter selectively kills proliferating cells while sparing post-mitotic neurons. Thus, we do not understand this comment of the referee.

Reviewer 3 Report

In the current manuscript by Locatelli and colleagues, they were searching for more effective suicide trigger genes to control unwanted PSC proliferation when using them for cell therapy, while at the same time to test different pyrimidine and guanosine analogs to induce the death of proliferative cells. The assays are well designed and data are solid. This study will benefit a wide range of audience in the stem cell technology development field but not just neural cell therapy researchers, since the safety of using PSC for cell therapy is priority issue to address. There is one concern that author didn’t checked is whether the insertion of SR39h genes and treatment with ACV, PCV, GCV will affect PSC differentiation process. At least couple of neurol linage markers should be checked to ensure the transduce of SR39h gene and treatment with these analogs won’t affect differentiation.

Minor point: The last couple of sentences (line 457-460) doesn’t seem like part of the manuscript, please revise.

Author Response

"There is one concern that author didn’t checked is whether the insertion of SR39h genes and treatment with ACV, PCV, GCV will affect PSC differentiation process. "

We did not observe any differences in differentiation capacity between wild type and SR39h-transduced neural cells (this statement is now added in the Method section). This assessment was based on beta3 tubulin immunostaining (Figure 4F). Given the time limitation by the editor (10 days) we were not able to add additional markers.

As the nucleoside analogs are given after differentiation, we did not see a compelling reason to investigate their impact on cellular differentiation.  Importantly for our project however, the nucleoside analogs did not have a deleterious effect on differentiated neurons. 

"Minor point: The last couple of sentences (line 457-460) doesn’t seem like part of the manuscript, please revise."

We corrected the manuscript and added figure number corresponding to these sentences:  (Supplementary Figure 4, small purple region indicated by the red arrow).

Round 2

Reviewer 1 Report

In this work Locatelli and colleagues developed a mechanism for controlling the proliferation of ePSCs.  After my last revision, the manuscript was improved and can be published after the authors address the following minor concerns:

1- I understand that DNA sequencing is not possible due to possible multi-target inserting of the gene in the cells genome.  Please add this limitation to the discussion section.

2 – Image 2 is duplicated in the new version.  Please choose the correct version.

3 – Please improve the overall resolution of supplementary figures 1, 2, 3 and 4.

Finally, I congratulate the authors for their work.

Author Response

1- I understand that DNA sequencing is not possible due to possible multi-target inserting of the gene in the cells genome.  Please add this limitation to the discussion section.

   We added a novel paragraph in the discussion concerning this point

2 – Image 2 is duplicated in the new version.  Please choose the correct version.

   This is now corrected

3 – Please improve the overall resolution of supplementary figures 1, 2, 3 and 4.

   On our side these figures appear to be high resolution, but we suspect that the loss of resolution is linked to the PDF format. We therefore now provide the figures also in word. 

Reviewer 2 Report

I don't see any data to address my question. So, I still doubt the effect in the real stem cell transplantation cases. In addition, the significance issue I raised in my initial report is also a concern.

Author Response

We performed additional modifications:

  • as requested by the referee we change the name of the cells from "embryonic pluripotent stem cells (ePSC)" to "embryonic stem cells (hESC)"
  • we have added data that transgene insertion have no impact on cellular differentiation potential (supplementary Figure. 4)

This manuscript is a resubmission of an earlier submission. The following is a list of the peer review reports and author responses from that submission.

Round 1

Reviewer 1 Report

In this work Locatelli and colleagues developed a mechanism for controlling the proliferation of ePSCs.  Such system makes of transfected TK and subsequent activation of nucleoside analogs inside the cell (NPCs and ePSCs), which interferes with DNA replication and leads to cell death.  The authors went far as using in vivo models to validate their system.  I found the manuscript well written, organized and supported with solid results.  Nevertheless, I believe the authors missed a key experiment in their study, and as such I must propose Major revisions to the manuscript.  Please consider the following:

Major:

1- The insertion of the plasmid containing Ki67-SR39h in the cells genome was not demonstrated in your work.  I believe this is especially important, since this plasmid differs from the one you used in your previous work.  I believe RT-PCR is not enough and does not account for the insertion/plasmid location inside the cell.  Please consider DNA-sequencing to prove this.

Minor:

Introduction

2- Please check if the word “investigated” in line 70 is adequate;

Results

3- In lines 97-98, please consider explaining why these sequences were chosen to the initial screening.  This information is further ahead in the text, but I feel it should be discussed early in the results section;

4- Please insert the number of independent experiments performed for the results presented in figures 2A (quantification available?) and 6A.

5- For Figure 4F, please increase the intensity of the green marker used.

6- The images of the ePSC nucleus are interesting results to show chromatin/DNA integrity.  Can you provide similar images for ReNcells?

Discussion

7- Regarding the differences observed between ACV, GCV and PCV, consider also comparing your results with their predicted affinity with TK.  Could this also explain your better results with PCV?

8- Regarding the in vivo assay, was animal behavior monitored throughout the experiment?  Can these results be provided?

Materials and Methods

9- References 4 and 37 are the same.  Please address this.

10- Please make a proper caption for the PCR primer table.

Reviewer 2 Report

The tumor-formation possibility is a critical challenge for stem-cell-based replacement therapy, and thus has attracted more attentions. In this manuscript entitled with “Optimization of thymidine kinase-based safety switch for neural cell therapy”, Locatelli and colleagues utilized the classical suicide gene system based on nucleoside analogs and thymidine kinase, together with genetic manipulation in human embryonic stem cells and neural progenitor cell line. They found the humanized SR39 (SR39h) thymidine kinase variant, together with the chemical Penciclovir (PCV) and ganciclovir (GCV), showing the best efficacy. In general, this study is too preliminary, and the conclusion is not well supported by the data.

One major concern is the in vivo transplantation. The study was aiming to solve the tumor-formation in neural cell therapy; however, the transplantation only tested the undifferentiated embryonic stem cells rather than the differentiation mixture.

Another major concern is the cell line/type bias. The authors tested one pluripotent cell line/type and one neural progenitor cell line/type, which is not enough to make conclusion that SR39h and PCV indeed work in stem cell transplantation.

In addition, the authors seem not familiar with stem cell biology. Embryonic stem cell is a specialized term and the most famous cell type of pluripotent stem cells. The term of “embryonic pluripotent stem cell” (ePSC) is hardly seen in literature.